# Cardiovascular Risk in Childhood Cancer Survivors

**DOI:** 10.3390/biomedicines10123098

**Published:** 2022-12-01

**Authors:** Francesca Mainieri, Cosimo Giannini, Francesco Chiarelli

**Affiliations:** Department of Paediatrics, University of Chieti, 66100 Chieti, Italy

**Keywords:** childhood cancer survivors, metabolic syndrome, diabetes, insulin resistance, hypertension, dyslipidemia, abdominal radiation, chemotherapy, metabolic derangements

## Abstract

Cancer is a prominent cause of death worldwide in the pediatric population. Since childhood cancer is not possible to prevent, it is essential to focus on a prompt and correct diagnosis followed by effective, evidence-based therapy with individualized supportive care. Given the enhancement of childhood cancer management over the past decades, survival rate has significantly improved, thus leading to the progression of several late effects, including metabolic derangements. These metabolic imbalances are associated with the underlying disease and the cancer treatments. As a result, the metabolic state may contribute to a high risk of cardiovascular morbidity and premature mortality among childhood cancer survivors. This review aims to summarize the potential pathophysiological mechanisms linked to the risk of diabetes and metabolic syndrome and screening recommendations. Further investigations are needed to clarify the underlying mechanisms of such metabolic abnormalities and to improve long-term cardiometabolic survival among these patients.

## 1. Introduction

Neoplastic diseases are the leading causes of death for children and adolescents. According to recent data, about 400,000 children and adolescents each year develop cancer [1,2]. Given the increasingly early diagnosis, the availability and administration of new specific and effective therapies has induced a significant improvement in prognosis. In fact, recent data has shown that children with neoplastic disease have significantly improved their survival rate childhood cancer over the last few decades. Specifically, annual survival rate for childhood cancer now exceeds 80% [3]. However, pediatric survivors show an elevated risk of health problems due to both their pathology and cancer therapy. Almost three-quarters of survivors will have a chronic health condition, with many of them presenting severe, disabling or life-threatening conditions. Furthermore, there is a strong possibility that survivors suffer from multiple conditions [4]. Particularly, current evidence suggest the development of pathological conditions leading to an increased cardiovascular risk in children who survived cancer [5,6]. Disorders, such as metabolic syndrome (MetS), and the combination of obesity, hypertension, dyslipidemia and insulin resistance (IR) have been considered as vigorous triggers of premature cardiovascular disease (CVD) later in life. Among these endocrinological late effects, the most common are MetS and diabetes mellitus (DM) [7]. It is also estimated that about 50% of survivors experience one or more hormonal imbalance during their lifetime [8]. Furthermore, CVD is the second leading cause of death in childhood cancer survivors (CCS) [9]. CCS are 7 times more likely to die from cardiac causes than the general population, as they experience a significantly high risk of developing CVD and related factors at young age [10]. Recent data report that 24-year-old CCS showed the same cumulative incidence of severe or disabling, life-threatening or fatal health conditions as their 50-year-old siblings [10]. Though, the risk of CVD development tends to vary and indeed depends on cancer type, children’s age, treatment exposure and patients’ personal factors [6]. The increasing survival rate in CCS has been then associated with other undesirable endocrine late effects, as shown in Table 1, that may present decades after the completion of therapy [11]. In this review our aim is to underline the risk of developing cardiometabolic conditions in CCS previously exposed to cancer treatment. Additional studies are therefore crucial to understand the pathophysiological mechanisms of metabolic complications and to ameliorate the quality and length of survival of these patients.

## 2. Metabolic Syndrome

MetS is characterized by a group of adverse metabolic conditions, including inflammatory and protrombotic states, that determine an increased risk for CVD. The latest consensus of the International Diabetes Federation, of the National Heart, Lung and Blood Institute and of the American Heart Association, in 2009, defined diagnostic criteria to diagnose MetS for adult patients [13,14]. Conversely, no univocal MetS guidelines are yet available for the diagnosis in pediatric patients. However, the definitions accessible now share the following criteria: central obesity, hypertension, hypertriglyceridemia, low HDL level, impaired glucose tolerance [15]. Each of the aforementioned component will be discussed below.

### 2.1. Diabetes and Insulin Resistance

Diabetes is a metabolic disease related to a significant rise of the risk of micro- and macro-vascular damage in children and adolescents, expected to increase in its prevalence worldwide in the future years. Type 1 diabetes mellitus (T1D) is likely the result of autoimmune-mediated pancreatic ß-cell destruction in genetically predisposed children. The resulting insulinopenia then leads to a state of hyperglycemia and the breakdown of insulin-dependent processes of cellular energy storage and synthesis [16,17,18]. In contrast, type 2 diabetes mellitus (T2D) is characterized by the presence of IR, which typically results in a compensatory increase in endogenous insulin production, subsequently followed by ß-cell failure [19]. Given the direct link between adiposity and the development of IR and T2D, survivors with adipose tissue accumulation are more likely to suffer from diabetes than survivors who do not present adiposity [20]. In addition to this, it must be emphasized that CCS have a higher risk of developing diabetes, particularly children who were treated at a younger age, as recently demonstrated [21,22]. Thus, long-term survivors show a higher risk of adverse health outcomes also due to the presence of DM. Metabolic late effects are most often observed in CCS who were previously exposed to radiation. Abdominal radiation, total body irradiation and exogenous corticosteroids are among the risk factors most likely to induce diabetes and CVD, as represented in Figure 1.

#### 2.1.1. Abdominal Radiation

Abdominal radiation is one of the main therapeutic interventions for solid tumors, such as neuroblastoma, Wilms tumor, different types of sarcomas and Hodgkin lymphoma. Teinturier et al. [23] and Cicognani et al. [24] published the first reports in 1995 and 1997, respectively, on the evidence of a higher risk of DM in patients who had received abdominal radiation as Wilms tumor therapy, suggesting a harming effect on pancreas due to radiation. The occurrence of DM can be explained by the damage to the tail of the pancreas caused by radiation, with the result of pancreatic insufficiency [25,26]. The prevalence of self-reported DM after 23.5 years of follow-up was 2.5% in survivors, caused by radiation, in particular abdominal (OR 2.7) and total body irradiation (OR 7.2), alkylating agents (OR 1.7) and younger age at diagnosis (OR 2.4), while cranial radiotherapy and corticosteroids did not show any link [27]. As further confirmation, the risk of DM development in cancer survivors exposed to abdominal radiation is increased 3.4-fold than in randomly selected siblings, as shown by an analysis of Childhood Cancer Survivor Study (CCSS), after adjusting for body mass index (BMI). Additionally, a difference has been registered between patients who had received radiation and those who had not, as the latter did not show an increased risk for DM. This result further established the predominant role of radiation therapy in the development of DM, instead of disease-specific factors [27]. The exact correlation between radiation dose and DM risk is not yet well established. As reported in the French–UK cancer survivor cohorts, in particular solid cancer or lymphoma survivors, the higher the radiation dose to the tail of the pancreas, the higher the DM risk, suggesting a direct relationship, then followed by a plateau in risk [25]. Although the evidence of an increased risk of DM with higher radiation doses was shown in a recent study of Hodgkin lymphoma survivors in the Netherlands, an evident plateau in risk has not been confirmed [26]. Together with the radiation dose, the area of irradiation affects the ultimate risk. In fact, the highest risk has been reported in survivors treated with ≥36 Gy to the para-aortic lymph nodes and spleen, as it includes the majority of the volume of the pancreas [28].

#### 2.1.2. Total Body Irradiation

As suggested by the definition, total body irradiation (TBI) is not focused on a specific body area, but it is addressed to the whole body. TBI is commonly used as a first step in high-risk onco-hematological patients, who subsequently need to undergo hematopoietic cell transplantation (HCT) [29,30]. Thus, TBI might cause also other endocrinopathies as a result of target organ damage due to radiation exposure, such as growth hormone deficiency (GHD) [31,32,33], thyroid dysfunction and hypogonadism [34,35]. TBI is recognized as an independent risk factor for DM, as revealed by a large cross-sectional study in almost 1000 adult survivors treated with HCT [36]. A study conducted by Neville et al. on 250 CCS confirmed TBI as an independent risk factor of DM, together with hypogonadism [37]. Hyperinsulinemia and IR are the main underlying pathophysiological mechanisms that lead to DM development related to TBI. It has been further shown that patients treated with TBI develop increased total adipose mass, with a critical adipokine profile represented by higher leptin and lower adiponectin concentrations [38,39]. This condition is consequentially associated with a reduction in lean body mass and muscle mass, even taking into consideration similar BMI to controls [40]. An altered distribution of fat mass, characterized by an augmented visceral adiposity and a decreased lean body mass, is also registered in both healthy individuals and CCS with GHD, determining at the end the presence of DM [41]. Generally, as shown by several studies, radiotherapy, both abdominal and total, is the most common independent risk factor of DM in large cohorts of CCS. De Vathaire et al. investigated the link with damage to the pancreas, underlying that the risk of DM is increased in a dose-dependent way because of radiation to the pancreatic tail, where the majority of insulin-secreting Langerhans islets is located [25]. An alternative pathologic mechanism is represented by the impaired fat cell expansion caused by radiotherapy that subsequently determine liver steatosis and circulation of free fatty acids (FFA), eventually leading to DM and IR [42].

#### 2.1.3. Exogenous Corticosteroids

It has been validated that CCS who received huge doses of exogenous corticosteroids during the years of disease can develop DM and metabolic consequences, including IR, obesity, osteoporosis and osteopenia, neurocognitive alteration and neuronal tissue damage [43,44,45]. This is particularly true for patients diagnosed with acute lymphoblastic leukemia (ALL). The pathophysiology of the onset of diabetes is multifactorial, but substantially lies in the increased IR and gluconeogenesis and decreased insulin production. A transitory condition of hyperglycemia in children while steroid-treating is registered and resolved just after therapy cessation [46,47]. However, the necessary long-lasting steroid therapy for this class of patients predisposes the progression to a permanent condition of diabetes [48]. Additionally, a permanent state of diabetes might be the consequence of the association of hyperglycemia with a sedentary life and irregular food intake [49]. However, a dose-response relationship between cumulative glucocorticoid dose and overweight in a large cohort of long-term CCS has not been found, according to recent data from the Swiss Childhood Cancer Study [50]. Production and secretion of insulin from pancreatic beta cells are influenced by dose, time of exposure and administration of corticosteroid treatment [51].

#### 2.1.4. Type 1 versus Type 2 Diabetes Mellitus Development

Although several studies have been conducted, it is not yet completely clear whether it is more common to develop T2D or T1D after cancer treatment [26,27]. According to the different pathophysiology of the two types of diabetes, T1D can be identified after the assessment of patients’ pancreatic autoantibody status, evaluation not realized by large studies. Nevertheless, a recent Scandinavian study of autoimmune disease among 20.361 1-year survivors of CCS showed that survivors were at a 1.6-fold increased risk for hospitalization due to insulin-dependent diabetes [52]. Comparably, the French–UK cohort described by de Vathaire et al. revealed a higher incidence of both insulin-dependent and insulin-independent diabetes after abdominal radiation [25]. In both cases, no information on autoantibody status was available. Therefore, a further study has been recently conducted by Friedman et al. to analyze the pancreatic autoantibody status. Autoimmunity resulted as a non-leading cause of post-therapy diabetes, since the presence of no more than one positive pancreatic autoantibody was registered, in the face of several glucose and insulin imbalances [53]. Additionally, the real metabolic situation of CCS needs an evaluation according to waist circumference, as BMI is not considered as a reliable marker to assess total body fat and fat distribution after radiation therapy [54]. The latest theories suggest that abdominal radiation provokes damages to the subcutaneous adipose tissue, determining lipid deposition in the visceral depots. This condition is then strictly related to the progression of chronic low-grade inflammation and metabolic disorders, finally including diabetes. So, there is a concrete chance to believe that the adipose tissue dysfunction produced by radiation, and not the obesity status, represents the main cause for the development of diabetes [28].

### 2.2. Obesity and Dyslipidemia

Overweight and obesity have become a central health issue over the past decades, especially in developed countries. Of note, alarming data has shown an increasing prevalence of increased adiposity in CCS. In particular, it has demonstrated that survivors of ALL have a high risk of becoming overweight or obese during the initial phase of the treatment, however maintaining the increased weight throughout treatment and beyond. As recently shown by Lindemulder et al., in a group of 269 standard-risk patients with ALL treated with cranial radiation therapy, the rate of overweight/obesity ranged from 14% at diagnosis to 39% at the end of therapy [55]. According to recent studies, young survivors, mostly preteenagers and adolescents who were off-treatment <10 years, had a substantially higher BMI than the reference population [56]. Furthermore, the prevalence of overweight/obesity at the end of treatment emerges to be higher than the prevalence in the general population [57]. Evidence suggests that CCS tend to eat a low-quality diet, made up of foods with high energy density, in addition to low expenditure because of a sedentary lifestyle. These two conditions have been reported among the predictors of obesity [58]. Greater depression and lower mobility were thought to be associated with greater fatigue and higher BMI in CCS [59]. Several complex pathways have been identified as responsible for obesity in patients who had previously received cancer treatments. Corticosteroids, especially when used for long periods, have shown to increase percentage of body fat in pediatric ALL survivors, also due to the significant increase in caloric intake that their administration tends to cause [60]. Cranial radiotherapy can directly damage the hypothalamic–pituitary region, impairing signaling reception from hormones, such as those involved in the regulation of appetite and hunger, such as ghrelin and leptin [61]. Some therapeutic agents can indirectly cause obesity, by determining physical impairments. For instance, in the case of cranial radiotherapy, obesity may be caused as a result of the decreasing muscle mass and strength and the impairment of balance and postural control, while anthracyclines may lead to obesity by determining left ventricular dysfunction and subsequent impairment in cardiovascular fitness [62,63]. Thus, obesity predisposes to the worsening of the elevated chronic health conditions already experienced by CCS. Therefore, the identification of risk factors for obesity is an essential step for improving the long-term health condition of these patients.

CCS, who present an elevated risk of overweight and obesity, carry an increased risk of dyslipidemia as well. This can be explained by the evidence that the release of FFA by the adipose tissue leads to increased triglyceride and very low-density lipoprotein (LDL) cholesterol production in the liver [64]. The resulting dyslipidemia, characterized by high fasting levels of total cholesterol and LDL cholesterol and triglycerides, and low levels of high-density lipoprotein cholesterol, is then linked to the progression of CVD [65]. Dyslipidemia can be also caused by hypogonadism, secondary to adult testicular cancer therapy [66,67]. In a study including 330 survivors evaluated after 16.1 years of follow-up, different conditions have been identified as independent risk factors. In particular, older age at diagnosis, GHD and autologous stem cell transplantation were independent risk factors for hypercholesterolemia, while those for hypertriglyceridemia were TBI and GHD [68].

### 2.3. Hypertension

The elevated risk of CVD among CCS, especially after the transition into adulthood, might also be a consequence of increased rates of hypertension in these patients [69]. Indeed, CCS are diagnosed with hypertension at a 2/3-fold higher rate than the general population. As reported by a Cochrane review by Knijnenburg [70], the prevalence of hypertension in these patients ranges from 0% to 18.2%, with an even higher prevalence registered thereafter, especially among older aged survivors, exceeding 70% by age 50 [69,71,72]. In addition, the combination of hypertension and radiotherapy further amplifies the cardiovascular risk [73]. However, not every survivor develops hypertension, but it has been proposed that only CCS who show a genetic predisposition go on to present hypertension. Thus, high blood pressure is hypothesized to be triggered by childhood cancer therapies. The underlying mechanisms able to cause hypertension involve a direct damage to the kidneys through irradiation and remaining kidney hyperfiltration due to unilateral nephrectomy [74,75]. Conversely, nephrotoxic effects by ifosfamide and cisplatin have not been reported to determine hypertension [72]. These inherited environment connections are particularly true for individuals with the highest inherited risk, further worsened with the advent of COVID-19 infection [76,77]. It is strongly suggested to precociously identify and treat subclinical hypertension in CCS by standard surveillance, as it decreases cardiovascular complications [69,72].

## 3. Cancer Treatment and Metabolic Derangements

Compared to siblings, CCS have shown a more elevated risk of MetS and a further higher risk of developing CVD [78]. As shown by the American Heart Association, researchers found that CCS were more likely than their peers to develop DM (6.5% versus 3.2%), dyslipidemia (14% versus 4.9%) and hypertension (18% versus 11%). It is worth to underline that both groups were just as likely to be underdiagnosed, as they were unaware of the conditions. Subsequently, CCS were almost twice as likely to receive inadequate treatment for these conditions (https://www.heart.org/en/news/2022/06/08/cardiovascular-risk-factors-undertreated-in-childhood-cancer-survivors, accessed on 8 June 2022). The underlying pathogenesis is still not completely known. However, chronic inflammation represented by increased C-reactive protein and cytokine activation (tumor necrosis factor-α, interleukin-6, etc.), released from abdominal adipose tissue as a consequence of treatment on organs and cardiovascular system, may have a role. Tumoral treatments, included radiotherapy, chemotherapy and long-lasting high-dose steroid therapy, affect the metabolic system producing imbalances and eventually facilitating the development of MetS [79].

### 3.1. Radiotherapy

Cranial radiotherapy is the main risk factor for hypothalamic–pituitary late effects, with anterior pituitary hormones, such as GH, LH/FSH, TSH and ACTH, being potentially affected, causing an important risk of secondary sequelae, as shown in Table 1 [8]. Hypothalamus/hypophysis axis is damaged by radiotherapy and this leads to the increase of the android/ginoide fat ratio with a subsequent central fat accumulation, responsible for the release of inflammatory molecules. When cranial dose radiation to hypothalamic–pituitary axis is high (>30 Gy), a leptin resistance on hypothalamic receptors occurs and higher circulation levels of leptin are documented. In fact, leptin resistance determines leptin overproduction by fat tissue [80,81], thus the lack of the central action of leptin causes an increase in fat tissue. Indeed, elevated blood leptin levels are related to glucose intolerance and IR and high BMI percentile for age and visceral adiposity. Particularly, CCS who present a condition of leptin resistance due to cranial radiation are then associated with higher BMI, fat mass and evidence of central adiposity [82]. Another factor resulting from cranial radiotherapy is the increase risk of progression of MetS onset is GHD. Indeed, this state is associated with high fasting insulin level, abdominal obesity and dyslipidemia, independently of radiation dose [83]. In addition, replacement therapy with GH further deteriorates IR and the induced high circadian GH levels, then tend to enhance lipid oxidation and free fatty acid production [84]. As a confirmatory step of the role of radiation, in a study of 532 adult long-term CCS, it has been demonstrated that decisive aspects causing risk of hypertension, increased waist circumference, DM and MetS are the treatment factors rather than genetic variation [85]. CCS of leukemia have been shown to present high prevalence of MetS, together with individual cardiovascular risk factors and obesity, with a higher prevalence for patients previously exposed to cranial radiation or TBI [86,87]. Additionally, as revealed from the French L.E.A. cohort, risk of MetS is highest among patients transplanted with TBI, followed by patients treated with chemotherapy and cranial radiation, transplantation without radiation and chemotherapy alone. Differences in the development of MetS based on the exposure history need to be reported, although further studies are requested to elucidate the underlying mechanisms [88].

TBI is considered another independent risk factor for the development of cardiovascular risk and MetS among CCS [89,90,91]. Interestingly, in survivors who are previously exposed to TBI, MetS can occur even in the absence of obesity. In large study, conducted in 2000 survivors who were on a 1-year HC, it has shown that those treated with TBI were found at greatest risk for the development of DM and dyslipidemia. Interestingly, the higher the number of cardiovascular risk factors, including hypertension, diabetes and dyslipidemia, the higher the 10-year incidence of CVD [92]. MetS is seen to develop also among other populations of cancer survivors and this is the case for neuroblastoma and Wilms survivors [54].

On the other hand, radiotherapy to other areas of the body can cause MetS as well. For instance, abdominal or chest radiation, together with steroid therapy, can determine an increase of both central systolic and diastolic pressure, probably due to the direct vascular damage and fibrosis [93].

### 3.2. Chemotherapy

The onset of MetS might be determined by the administration of chemotherapeutic agents. Although these drugs perform direct and indirect actions, it is challenging to highlight the role of each chemotherapeutic agent in determining MetS, as they are often administered together. Some chemotherapeutic agents, by exerting their harmful action, have the cardiovascular system as their principal target. Therefore, cardiotoxicity is induced by therapy with anthracyclines, which cause cardiovascular damage and hypertension. Therefore, anthracycline-related cardiotoxicity produces left ventricular pathological remodeling, fibrosis and afterload abnormalities [94]. Conversely, doxorubicin acts inducing p53 expression, thus leading to cardiomyocyte atrophy and death and a decrease in heart weight [95]. Platinum therapy is more often associated with the development of IR and cardiovascular risk in CCS, due to endothelium injury after cytokines release [96,97]. L-asparaginase and steroids are administered to patients with acute lymphoblastic leukemia (ALL) during the induction therapy, inducing hyperglycemia. L-asparaginase action consists of assessing a systemic insulinopenic state and hyperglucagonemia by directly inhibiting insulin production and decreasing insulin receptors. Insulin production is further indirectly inhibited by L-asparaginase by the induction of pancreatitis which determine beta cell damage [98]. The risk of MetS development is particularly elevated in ALL patients during puberty and this can be more likely explained by the fact that sex hormone excretion might enhance glucose intolerance and IR. Additionally, cytokine production, anti-inflammatory dysfunction and increased reactive oxygen species, regarding gut environment, strongly contribute to the onset of MetS [99,100].

### 3.3. Steroid Therapy and Lifestyle Factors

As previously described, steroid therapy is involved in the pathogenesis of MetS as it causes a great risk of developing DM with significant fat accumulation, cytokine release and chronic inflammation, which finally determines DNA damage. This pathological vital organ failure is able to determine relevant consequences, such as liver steatosis, precocious atherosclerosis, thrombosis and myocardial infarction [43].

A heart-healthy lifestyle, based on physical activity, diet and no smoking habits, is essential to avoid the development of MetS for both in the general population and CCS. A recent study on 1598 adult CCS has demonstrated that patients who did not follow World Cancer Research Fund/American Institute for Cancer Research (WCRF/AICR) recommendations for a healthy lifestyle were at a greater risk to develop criteria for MetS, when compared to those who followed these guidelines [78]. Moreover, further studies have revealed that an amelioration of cardiometabolic risk factor status is seen when CCS practice physical activity, especially HCT survivors [101,102,103]. This underlines how influential lifestyle changes can be on the metabolic structure of these patients. However, data in these subjects need further studies to provide additional and relevant results in the field.

## 4. Cardiotoxicity and Cardiomyopathy

Chest radiotherapy, often used in children with the presence of lymphoma, has been demonstrated to raise the risk of several cardiovascular conditions, such as coronary artery disease, pericardial disease, valvular and conduction system disease and cardiomyopathy [104]. These cardiac pathological conditions caused by radiation occur with an estimated incidence of 10–30%. Additionally, specific treatments of childhood cancer are cardiotoxic, especially anthracyclines and tyrosine kinase inhibitors [105]. Doxorubicin, daunorubicin and epirubicin, all belonging to the anthracyclines class, are often used to treat both hematologic and solid malignancies, and has shown great outcomes in patients with ALL and sarcomas [106,107]. However, anthracyclines determine several adverse effects and it is clearly shown from the increasing cardiotoxicity registered in about half of all CCS within 20 years of receiving anthracyclines [108,109]. It is thought that anthracyclines cause a dose-dependent and irreversible loss of cardiomyocytes, with a cumulative lifetime dose of more than 300 mg/m^2^ imparting the highest risk [110,111]. Furthermore, a decreased left ventricular fractional shortening may be the result of a reduced left ventricular wall thickness and mass due to anthracyclines therapy [112,113]. During childhood, anthracyclines-associated cardiotoxicity consists of an asymptomatic dilated cardiomyopathy, often reversible after cessation of anthracycline therapy. Instead, results obtained by conducting an extended follow-up, many CCS who had been treated with anthracyclines have shown a restrictive cardiomyopathy with reduced left ventricular wall thickness and left ventricular mass for body-surface area and increased left ventricular wall stress (afterload), which drives downregulation of left ventricular systolic function due to left ventricular diastolic dysfunction [113]. The condition wherein the left ventricular mass is too small for the bodies of these long-term anthracycline-treated CCS has been termed as the “Grinch Syndrome”. Namely, it consists of a chronic cardiomyopathy, which may result in heart failure, heart transplantation, or premature death in long-term survivors [114]. Cardiotoxicity may also result from therapy with other chemotherapeutic and targeted drug therapies that may eventually be used to treat children. Particularly, alkylating agents, such as ifosfamide and cyclophosphamide, may cause heart failure, the antimetabolite fluorouracil may lead to cardiac ischemia and bevacizumab has been associated with myocardial infarction [115,116]. The onset of these cardiotoxic effects is categorized as acute, early or late, depending on the time since anthracycline administration, as shown by Lipshultz and Adams [108,117]. Risk factors for severe cardiotoxicity include higher lifetime cumulative doses of anthracyclines, higher anthracyclines dose rates, younger age at treatment, longer follow-up after anthracyclines treatment, female sex and cardiac irradiation [114].

The underlying mechanisms of anthracycline-induced cardiotoxicity are complex, but one of the acknowledged mechanisms is the “oxidative stress hypothesis”, which leads to cellular damage and death [118,119]. Several other mechanisms of cardiotoxicity include the up-regulation of nitric oxide synthetase and the alteration of gene expression, determining impaired creatine kinase activity and function in mitochondria [120]. Some of these cellular sequelae progress for weeks after therapy with anthracyclines, clarifying some mechanisms of chronic cardiomyopathy [121]. Thus, the long-term anthracyclines-induced cardiotoxicity is a major limitation of the use of these chemotherapy agents, with a progressively higher rate when an extended follow-up is realized [111]. However, newer protocols limit the chemotherapy dosing and improve radiation accuracy, which has shown to decrease the acute clinically evident cardiovascular complications to less than 1% [122]. A greater risk has been registered in children who were facing pre-existing cardiovascular risk factors, hence the importance of early recognition of heart disease in cancer survivors [123]. It is estimated that as many one in eight CCS treated with anthracyclines and chest radiotherapy will experience a life-threatening cardiovascular event 30 years after the treatment of childhood cancer [124]. In fact, the American Society of Echocardiography (ASE) and European Association of Cardiovascular Imaging (EACVI) Expert Consensus recommend a screening of the cardiac function for all patients prior to receiving cardiotoxic agents. This is particularly true for all those patients at higher risk, as shown in Table 2 [125].

## 5. Vascular Abnormalities

In addition to CVD, recent studies have shown that CCS later in life usually experience the development of premature arterial disease, due to the use of both radiotherapy and some chemotherapy. Increased carotid intima-media thickness (IMT), stiffer arteries and endothelial dysfunction are among the most reported vascular abnormalities [126,127]. All these vascular anomalies may be considered as predictable of early atherosclerosis and markers of later onset of ventricular dysfunction, stroke and ischemic heart disease [128,129]. Especially irradiation to the mediastinum, head or neck, affects the IMT, particularly in males. This suggests that male gender is an independent risk factor predisposing irradiated male to develop atherosclerosis. However, it has been demonstrated that female patients are more predisposed to develop different late effects, such as cardiovascular late effects or MetS [130]. Moreover, anomalies in the IMT were more visible in the irradiated group than in the patients treated only with chemotherapy, suggesting that irradiation, event at lower or moderate doses, is the main factor predisposing to early development of atherosclerotic vascular disease in CCS [126]. IMT is assessed by vascular ultrasound, that is a non-invasive technique able to detect early preclinical sonographic evidence of atherosclerosis. In particular, a higher IMT in major part of the areas studied in survivors previously treated with anticancer therapy has been registered, especially in those irradiated, as compared with the control. Often, the first signs of carotid IMT show up >1 year after the end of treatment [131,132]. Conversely, the highest rate of stroke was noted in the group treated for brain tumor with irradiation doses > 50 Gy [133]. Vascular structure can be damaged by radiotherapy, which often causes morphological changes, ischemic lesions in the arterial wall due to vasa vasorum injury, loss of elasticity, fibrosis and all this situation predisposes to chronic inflammation and endothelial cell dysfunction [134]. The assessment of the early onset of vascular changes, together with the cardiac alterations, may help to discover the development of CVD in CCS and facilitate new and better preventive strategies

## 6. Lifelong Screening, Long-Term Follow-Up and Treatment

Screening timeline represents a key component of the follow up of CCS (Table 1). Children’s Oncology Group (COG) guidelines suggest that survivors exposed to abdominal radiation or TBI should have a fasting lipid profile, fasting blood glucose or hemoglobin A1c checked every 2 years or more often if needed [12]. In addition, individuals with a positive family history of T2D are at higher risk due to genetic predisposition, thus closer surveillance must be ensured [135]. A fundamental step for prevention in CCS is practicing regular physical activity and following a heart-healthy diet, since the resulting outcomes have been registered as beneficial [136]. On the other hand, COG guidelines removed any specific screening guidelines for MetS linked to cranial radiation or TBI. However, height, weight and blood pressure should be checked annually in these patients, with assessment of nutritional status [12]. Further strategies for prevention or treatment of MetS in CCS are still lacking. Despite existing guidelines that can be utilized for primary prevention of cardiometabolic complications, a modified strategy and an individual based approach to minimize the progression of metabolic abnormalities is suggested [137]. Given the high prevalence of metabolic imbalances in CCS, a definite life-long care based on the most common risks is necessary. It is imperative to follow these patients over the years, as the effects of the therapy may occur after a long period of clinical latency. For instance, this is the case of the link between abdominal radiation and the development of DM [25]. Regarding CVD, many survivorship guidelines have recommended risk-based cardiomyopathy surveillance beginning after the completion of cancer therapy, allowing early detection and treatment of asymptomatic cardiomyopathy [138]. Echocardiography has become the preferred screening modality due to relatively low cost, acceptable sensitivity and specificity and widespread availability [139]. More recently, serum biomarkers of cardiotoxicity (cTnT, cardiac troponin I and N-terminal pro-brain natriuretic peptide) are increasingly being used to screen CCS for cardiotoxicity caused both years later and during anthracyclines therapy [94,140]. Interestingly, the two leading causes of morbidity and mortality, namely cancer and CVD, have shown to share biological mechanisms. Inflammation and oxidative stress are considered the common link involved in these processes. Additionally, several risk factors, such as obesity, tobacco abuse, sedentary lifestyle and imbalanced caloric intake have seen to cause both neoplastic and heart disease [141]. The earlier a correct screening is realized, the earlier the treatment-related complications can be identified. An endocrinological consultation is strongly recommended since different metabolic alterations can be well-managed by endocrine hormone replacement. However, data on survivor-specific treatment recommendations for metabolic abnormalities in these high-risk patients are still lacking. In addition, MetS cannot be treated by a single drug therapy [142]. These conditions further strengthen the need to develop specific, efficient and early preventive strategies and follow-up screening to avoid the development of metabolic dysfunctions, and novel therapeutic approaches capable of improving the overall survival of this population.

## 7. Conclusions

As shown so far, CCS present several late effects due to the treatments received, including metabolic consequences. CVD, GHD, DM and MetS are some of the most frequent risks these patients can develop. It is clear that some survivors with specific medical history and genetic characteristics are more likely to present metabolic complications. Specific risk factors for the development of such derangements have been identified, including cranial and abdominal radiation, TBI, chemotherapy and exogenous steroids; however, the underlying exact mechanisms are not well-defined so far. As these patients are already at increased risk for early cardiovascular morbidity and mortality, a primary screening and realization of several risk-reducing approaches are firmly indicated. Therefore, further studies are necessary to explain the pathophysiology of these metabolic conditions, in order to develop preventive interventions and novel therapeutic strategies and improve the survival in terms of quality and duration of life.

## Figures and Tables

**Figure 1 biomedicines-10-03098-f001:**
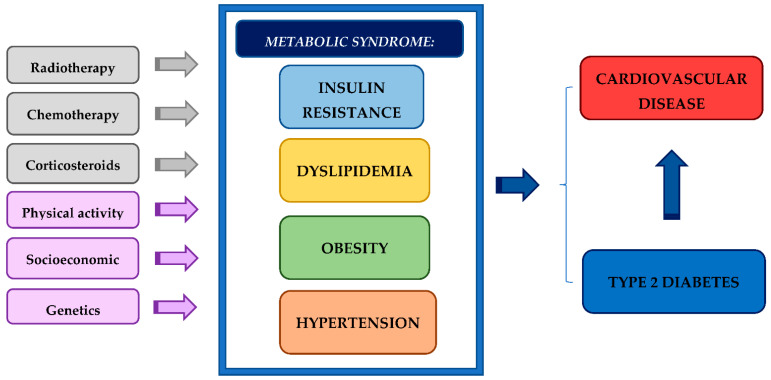
Treatment and environmental factors of the component of metabolic syndrome in childhood cancer survivors and the resulting risk of developing cardiovascular disease and type 2 diabetes.

**Table 1 biomedicines-10-03098-t001:** Endocrine late effects due to risk factors related to cancer treatment per Children’s Oncology Group [11,12].

*Endocrine Late Effects*	Treatment Risk Factors	Timing of Screening
*GHD*	Radiation (head/brain), TBI)	Every 6 months until growth complete, then yearly Every 6 months until sexually mature
*ACTH deficiency*	Radiation (head/brain)	Yearly
*LH/FSH deficiency*	Radiation (head/brain), TBI)	Yearly
*TSH deficiency*	Radiation (head/brain)	Yearly
*ADH deficiency*	Brain surgery	Yearly
*Hyperprolactinemia*	Radiation (head/brain)	Yearly
*CPP*	Radiation (head/brain)	Yearly, until sexually mature
*Non-GHD short stature*	Any radiation (including TBI)	Yearly
*Primary hypothyroidism, Hyperthyroidism, thyroid nodules/cancer*	Radiation (head/brain, neck, cervical/whole spine, TBI) * only for primary hypothyroidism: MIBG, thyroidectomy	Yearly
*Female gonadal dysfunction*	Traditional alkylators, heavy metals, radiation (pelvis, sacral/whole spine, TBI), oophorectomy	Yearly
*Male gonadal dysfunction*	Traditional alkylators, heavy metals, radiation (testicular, TBI), orchiectomy	Yearly

*: MIBG and thyroidectomy are treatment risk factors for primary hypothyroidism only, but not for hyperthyroidism and thyroid nodules/cancer. GHD: growth hormone deficiency; ACTH: adrenocorticotropic hormone; LH: luteinizing hormone; FSH: follicle-stimulating hormone; TSH: thyroid-stimulating hormone; ADH: antidiuretic hormone; CPP: central precocious puberty; TBI: Total Body Irradiation.

**Table 2 biomedicines-10-03098-t002:** Pathological conditions leading CCS to higher risk of subsequent development of CVD after radiotherapy.

Pre-Existing Risk Factors of CVD
Prior LV dysfunction
Structural heart disease
Family history of CVD
Age > 65 years old
Treatment with high dose of anthracycline alone or in combination with other cardiotoxic chemotherapy

## Data Availability

Not applicable.

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
