# Peer review of "Cardiovascular Risk in Childhood Cancer Survivors"

_biomedicines, 2022, doi:10.3390/biomedicines10123098_

Round 1
Reviewer 1 Report
This is quite appreciable that the review has been covered the risk of cardiovascular morbidity and mortality among childhood cancer survivors pretty good. Although there are some main issues, which need to address:
1) One most important section: “Cardiomyopathy” is missing. “Cardiomyopathy” should be sectioned separately rather than diluting into whole manuscript to make the manuscript attractive. Without this section the title mentioned “Cardiovascular risk in childhood cancer survivors” will better replaced with “Metabolic risk in childhood cancer survivors”
2) It will be noteworthy to include vascular anomaly related pathology among cancer survivors too.
3) It would be appreciable to put following the sections: “Diabetes and Insulin Resistance”, “Dyslipidemia” and “Hypertension” under the section of “Metabolic syndrome”, which is nicely portrayed in figure 1. Otherwise, it would be better to remove the section “Metabolic syndrome”.
4) “Obesity” section is missing, which should include as separate section or along with “Dyslipidemia” section.
5) Unfortunately, molecular pathways are less discussed or highlighted in each section of the manuscript, which would strengthen the manuscript to the audience.
There are few minor issues to highlight:
1) There are many grammatical errors in the manuscript, e.g.
· On line 22, “Neoplastic disease is a leading cause…...” will be “Neoplastic diseases are the leading causes…..”
· Kindly rewrite the 2nd part of 1st sentence; “designed to intensely grow world-54 wide in the future years.” on section “2. Diabetes and Insulin Resistance”.
· On line 64-65, “………. were treated at a young age” will be “………. were treated at younger age”.
· On line 65-66 please rephrase the sentence: “The itself presence of DM will increase……... own risk of adverse health outcomes”.
· I guess on line 69 you wanted to write the sentence “…...likely risk factors to induce diabetes, cardiovascular diseases, as represented in Figure 1” than “……...likely to cause risk factors, including diabetes, as represented in Figure 1”. Because radiations, steroids are the risk factors. If not, please rephrase the sentence.
· On line 87-88, “Teinturier et al. first [19] and Cicognani et al. after [20], respectively in 1995 and 1997, published the first reports on the…….” should write as “Teinturier et al. [19] and Cicognani et al. [20] published the first reports in 1995 and 1997, respectively on the…….”
2) Please give attention to mention all the abbreviations. For example, on table 1 ACTH, LH/FSH, TSH, ADH are not abbreviated anywhere.
3) It would be better to replace the oldest references (such as ref. no. 19, 20, 28, 36.) with the latest ones.
I would recommend revising the manuscript by an English-speaking person.
Thanks again for your wonderful effort to make an interesting manuscript.
Reviewer 2 Report
The authors reviewed cardiovascular risk in childhood cancer survivors. However, I have some comments.
1) The authors described the higher prevalence of diabetes mellitus (DM), metabolic syndrome, dyslipidemia and hypertension in childhood cancer survivors. However, I would like to know whether or not the prevalence of cardiovascular disease in such childhood cancer survivors is absolutely higher than in subjects without childhood cancer. When will most childhood cancer survivors develop cardiovascular disease?
2) The authors described screening and follow-up of DM, metabolic syndrome, dyslipidemia and hypertension in childhood cancer survivors. However, please describe briefly how to treat DM, metabolic syndrome, dyslipidemia and hypertension in such childhood cancer survivors. Is the treatment in such patients similar to that in subjects without childhood cancer?
3) A thorough review by a native English speaker would be helpful.
Round 2
Reviewer 1 Report
Dear Authors,
This nice review is looking quite better now. But still there are few minor issues to highlight:
· On line 196, “…...adiposity also in CCS” will be “….. adiposity in CCS”.
· On line 224, “…...the long-term health of these patients” will be “….. the long-term health condition of these patients”.
· On line 256, “Compared to their siblings……….” will be “Compared to the siblings …..”.
· On line 372, “………. drives left ventricular systolic function down due to left……….” will be “……….drives downregulation of left ventricular systolic function due to left……….”.
· What do mean by “left ventricular is too small” on line 373-374. Kindly rewrite the sentence: “The condition wherein the left ventricular is too small……..”.
· On line 417-18, “………. an independent risk factor predisposing irradiated males to develop atherosclerosis” will be “…….an independent risk factor predisposing irradiated male to develop atherosclerosis.”
· On line 456-57, “……beginning after completion of cancer therapy, allowing for early detection and treatment of asymptomatic cardiomyopathy” will be “…….beginning after the completion of cancer therapy, allowing early detection and treatment of asymptomatic cardiomyopathy”.
· On line 458, “Echocardiography has been the preferred screening modality, as it relatively low……” will be “Echocardiography has become the preferred screening modality due to relatively low …….”.
Thanks for your wonderful effort to make an interesting manuscript.
Author Response
Dear Reviewer,
Thank you very much for your precious suggestions.
All the highlighted minor issues have been corrected, as you can see in the updated version of the manuscript.
Thanks again.